# A Smart Multi-Rate Data Fusion Method for Displacement Reconstruction of Beam Structures

**DOI:** 10.3390/s22093167

**Published:** 2022-04-20

**Authors:** Qing Zhang, Xing Fu, Zhiguo Sun, Liang Ren

**Affiliations:** 1State Key Laboratory of Coastal and Offshore Engineering, Dalian University of Technology, Dalian 116023, China; zhang37qing@mail.dlut.edu.cn (Q.Z.); renliang@dlut.edu.cn (L.R.); 2Key Laboratory of Building Collapse Mechanism and Disaster Prevention, China Earthquake Administration, Langfang 065201, China; sunzhiguo@cidp.edu.cn

**Keywords:** structural health monitoring, multi-rate data fusion, dynamic displacement reconstruction, strain mode shape, Kalman filtering algorithm

## Abstract

Dynamic displacement plays an essential role in structural health monitoring. To overcome the shortcomings of displacement measured directly, such as installation difficulty of monitoring devices, this paper proposes a smart reconstruction method, which can realize real-time intelligent online reconstruction of structural displacement. Unlike the existing approaches, the proposed algorithm combines the improved mode superposition methods that is suitable for complex beam structures with the Kalman filtering approach using acceleration and strain data. The effectiveness of the proposed multi-rate data fusion method for dynamic displacement reconstruction is demonstrated by both numerical simulation and model vibration experiment. Parametric analysis shows that the reconstruction error is only 5% when the noise signal to noise ratio is 5 dB, illustrating that the proposed algorithm has excellent anti-noise performance. The results also indicate that both the high-frequency and low-frequency components of the dynamic displacements can be accurately reconstructed through the proposed method, which has good robustness.

## 1. Introduction

Nowadays, more and more large-scale bridge structures appear in human society. These structures are usually designed with greater flexibility; consequentially, the dynamic response is large under intense wind loads and earthquake loads. The dynamic displacement is the intuitive performance of the structure under environmental loads [1,2,3], which plays an important role in the field of bridge health monitoring since it contains structural deformation information [4,5,6].

In the past few decades, the research on structural dynamic displacement measurement has made great progress. The equipment used to measure the displacement of actual engineering structures directly includes global positioning systems (GPS), linear variable differential transformers (LVDT), laser Doppler vibrometers (LDV), and vision-based systems [7,8,9,10]. However, GPS has difficulty meeting the measurement requirements of all structural displacements due to the low sampling rate and poor accuracy. The application of contact sensors such as LVDT requires the installation of a fixed reference point, and the measurement results are also greatly affected by the support. The vision-based approach requires good visual conditions to be able to maximize its effect and LDV is difficult to employ in the full-scale measurement of structures due to its measurement characteristics. All these indicate that it is challenging to directly measure the displacement of real structures.

Therefore, the indirect methods of displacement reconstruction are studied [11,12]. The common approach is to use some easily obtained dynamic response to invert the dynamic displacement [13,14,15], which includes double integration of acceleration to obtain displacement time history. Nevertheless, the acceleration integration process often has an integration error, and the error will gradually accumulate with the integration process, and it cannot correctly reconstruct pseudo-static displacement or non-zero mean displacement. Except for acceleration, the strain is also used to reconstruct displacement since it is easy to measure and has a particular relationship with displacement [16,17,18]. In contrast, a lower strain sampling rate means that the strain does not include the information of the high-frequency components of the displacement. In other words, the strain at this time can only reconstruct the low-frequency displacement.

To overcome the limitation of using a single type of data such as acceleration or strain to reconstruct the displacement, researchers try to introduce Kalman filtering into the work of dynamic displacement reconstruction to achieve the best estimation of displacement [19,20,21]. Nevertheless, most of the existing Kalman filtering algorithms use acceleration and displacement or velocity as state variables, whose practical applicability is greatly limited. The displacement reconstruction method of bridge structures using easily collected data is needed.

The above results indicate that experts have invested a lot of energy into studying the dynamic displacement reconstruction of the bridge structure, which provided many ideas and references. Whereas, the equipment used for dynamic displacement measurement still has many issues, such as double integration and filter correction methods that cannot accurately reconstruct low-frequency displacement. The existing mode superposition method is not suitable for complex beam structures. Therefore, this paper proposes a smart multi-rate data fusion method, which combines an improved mode superposition approach to derive low sampling rate displacement online; afterwards, the acceleration and strain-derived displacement are used as state variables in the Kalman filtering approach to reconstruct high accuracy and sampling rate displacement in real time. Then, the practicability of the proposed method for complex beam structures is verified by the numerical simulation, as well as by the model experiment and parameter analysis. Finally, the conclusions and summary are given.

## 2. Smart Multi-Rate Data Fusion Method Using Acceleration and Strain

### 2.1. Basic Theory of Mode Superposition and Multi-Rate Kalman Filtering

#### 2.1.1. Mode Shape Superposition Method

For a structure with *r* strain measurement points, considering the first *n*-order strain modes, the strain response can be expressed as
(1)$${\left\{E\right\}}_{r\times 1}={\left[\Psi \right]}_{r\times n}\cdot {\left\{\eta \right\}}_{n\times 1}$$
where ${\left\{E\right\}}_{r\times 1}$ represents the strain response vector of the structure; ${\left[\Psi \right]}_{r\times n}$ stands for the strain mode matrix of the structure; ${\left\{\eta \right\}}_{n\times 1}$ denotes the modal coordinate vector of the structure.

According to the mode superposition theory, the strain and displacement response of a linear structure under environmental load have the same modal coordinates, which can be expressed as
(2)$$u(x,t)=\sum_{i=1}^{n}\Phi_{i}(x)\eta_{i}(t)$$
(3)$$\varepsilon (x,t)=\sum_{i=1}^{n}\Psi_{i}(x)\eta_{i}(t)$$
where the u(x,t) and ε(x,t) represent the displacement and strain response of position *x* at time *t*, respectively; $\Phi_{i}(x)$ and $\Psi_{i}(x)$ stand for the modal displacement of the *i*th order displacement mode and strain mode at the position *x*, respectively; *n* indicates the mode order used; $\eta_{i}(t)$ represents the *i*th modal coordinate.

After obtaining the strain mode shape $\Psi_{i}(x)$ of the structure, the least square method can be used to solve the modal coordinates ${\left\{\eta \right\}}_{n\times 1}$:(4)$${\left\{\eta \right\}}_{n\times 1}={\left({\left[\Psi \right]}_{r\times n}^{T}\cdot {\left[\Psi \right]}_{r\times n}\right)}^{-1}\cdot {\left[\Psi \right]}_{r\times n}^{T}\cdot {\left\{E\right\}}_{r\times 1}$$

Furthermore, from the beam bending theory of material mechanics, for the simply supported beam structure:(5)$$\varepsilon (x,t)=-y\cdot \frac{\partial^{2}u(x,t)}{\partial x^{2}}=-y\cdot \sum_{i=1}^{n}{\Phi^{\prime \prime }}_{i}(x)\eta_{i}(t)$$
where *y* represents the distance from the measurement point to the neutral layer of the structure. The position of the neutral layer of the actual bridge structure is not often in its geometric center, so calibration experiments are needed to determine the distance from the strain measurement point to the neutral layer. Two strain sensors are arranged on the upper and lower sides of the mid-span position at the bridge, respectively, which are correspondingly denoted as $s_{1}$ and $s_{2}$. Assuming that the total thickness of the bridge is *h*, and the distance y from $s_{1}$ sensor to the neutral layer is:(6)$$y=\frac{\left|\varepsilon_{1}\right|}{\left|\varepsilon_{1}\right|+\left|\varepsilon_{2}\right|}\cdot h$$
where $\varepsilon_{1}$ and $\varepsilon_{2}$ are the strain measurement values of sensors $s_{1}$ and $s_{2}$, respectively.

Combining Equations (3) and (5), the *i*th order displacement mode function can be obtained:(7)$$\Phi_{i}(x)=-\frac{1}{y}\cdot (\iint \Psi_{i}(x)dx^{2})+Cx+D$$
where *C* and *D* are two integral constants, which can be determined by the boundary conditions of the structure. The two integral constants of a simply supported beam are:(8)$$C=\frac{{\left.\frac{1}{y}\cdot \iint \Psi_{i}(x)dx^{2}\right|}_{x=l}}{l}$$
(9)D=0
where *l* denotes the full length of the simply supported beam. After calculating the first *n*-order displacement mode shapes, composing them into a displacement mode matrix:(10)$$\left\{\Phi (x)\}=\{\Phi_{1}(x),\Phi_{2}(x),\cdots ,\Phi_{i}(x),\cdots ,\Phi_{n}(x)\right\}$$
where Φ(x) represents the modal displacement value of the displacement mode matrix at position *x*.

Substituting the modal coordinates ${\left\{\eta \right\}}_{n\times 1}$ calculated by Equation (4) and displacement mode matrix Φ(x) calculated by Equation (10) into Equation (2), the displacement response u(x,t) can be obtained.

Considering that the non-uniform beam or continuous beam structures shown in Figure 1 are more likely to be encountered in actual engineering, the above mode shape superposition method needs to be improved. For a continuous beam structure, it can be regarded as consisting of independent simply supported beams since the displacement value at the hinge support is always 0. For example, the continuous beam in Figure 1a can be seen as consisting of two simply supported beams. Therefore, the sensors only need to be arranged on the target span and the mode superposition method can be used to solve the strain-derived displacement.

For the simply supported beam with non-uniform cross-section in Figure 1b, the mode shape superposition method cannot be directly applied to the entire structure due to the sudden change in the cross-sectional size. After the stochastic subspace identification (SSI) method is utilized to obtain the strain mode displacement values of all measurement points, the structure can be regarded as two equal cross-section beams to fit the strain mode function, respectively. Hence, the displacement mode function can also be expressed in sections as:(11)$$\Phi_{i}(x)=\left\{\begin{array}{l} \Phi_{i}^{l}(x)=-\frac{1}{y_{l}}\cdot (\iint \Psi_{i}^{l}(x)dx^{2})+C_{l}x+D_{l}\;x<x_{0}\\ \Phi_{i}^{r}(x)=-\frac{1}{y_{r}}\cdot (\iint \Psi_{i}^{r}(x)dx^{2})+C_{r}x+D_{r}\;x\geq x_{0} \end{array}\right.$$
where $\Phi_{i}^{l}(x),\Phi_{i}^{r}(x)$ are the displacement mode functions of the left section and right section of the beams, respectively; $\Psi_{i}^{l}(x)$ and $\Psi_{i}^{r}(x)$ are the strain mode functions of the left section and right section of the beams, respectively; $C_{l},C_{r},D_{l},D_{r}$ are all integral constants; $y_{l},y_{r}$ are the distances from the strain measurement point to the neutral layer of the respective beam, respectively. Based on the continuity condition, the left and right sides of the abrupt cross-section have the same rotation angle and vertical displacement, and by the boundary conditions at both ends, there are:(12)$$\begin{array}{l} \Phi_{i}^{l}(0)=0\;\Phi_{i}^{r}(L)=0\\ \Phi_{i}^{l}(x_{0})=\Phi_{i}^{r}(x_{0})\\ {\left.\Phi_{i}^{l}{}^{\prime }(x)\right|}_{x=x_{0}}={\left.\Phi_{i}^{r}{}^{\prime }(x)\right|}_{x=x_{0}} \end{array}$$

According to the above formulas, all the integral constants can be obtained, and then the two displacement mode functions can be solved. The subsequent calculation process for solving the mode coordinate and strain-derived displacement is consistent with the aforementioned mode shape superposition method. Once the modal information is identified, the corresponding displacement value can be quickly calculated for each strain value, realizing the real-time reconstruction of the displacement, so this method is a smart reconstruction algorithm.

#### 2.1.2. Multi-Rate Kalman Filtering Technique

Considering the state variables of the Kalman filtering continuous-time state space model are:(13)$$X(t)={\left[\begin{array}{cc} x(t) & \dot{x}(t) \end{array}\right]}^{T}$$
where x(t) is displacement, $\dot{x}(t)$ indicates velocity.

Considering the differential form of the state variable, the continuous-time state space model can be expressed as
(14)$$\dot{X}(t)=\left[\begin{array}{cc} 0 & 1\\ 0 & 0 \end{array}\right]X(t)+\left[\begin{array}{c} 0\\ 1 \end{array}\right]\ddot{x}(t)+\left[\begin{array}{c} 0\\ 1 \end{array}\right]\theta (t)=A_{d}X(t)+B_{d}X(t)+\theta$$

Then the discrete state space model can be expressed as
(15)$$X(k)=AX(k-1)+B\ddot{x}(k-1)+\theta (k-1)$$
where A represents the discrete-time state transition matrix; B denotes the control matrix; the calculation formula is
(16)$$A=\left[\begin{array}{cc} 1 & \Delta t\\ 0 & 1 \end{array}\right],\;B=\left[\begin{array}{c} 0.5\Delta t^{2}\\ \Delta t \end{array}\right],\;\theta (k-1)=\left[\begin{array}{c} w(k-1)\\ v(k-1) \end{array}\right]$$
where w(k−1) and v(k−1) represent noise related to acceleration and velocity, respectively. Assuming it is zero mean Gaussian white noise with variances *q* and *r*, respectively; Δt denotes the sampling interval. The one-step state vector prediction of the discrete-time state space model is:(17)$$X(k\left|k-1\right.)=AX(k-1\left|k-1\right.)+B\ddot{x}(k-1)$$
where $X(k-1\left|k-1\right.)$ means the state vector at k−1 time step, $X(k\left|k-1\right.)$ represents the state vector at the k time step predicted by the state vector at the k−1 time step.

The predicted state vector has errors, and the covariance matrix of one-step state prediction is
(18)$$P(k\left|k-1\right.)=AP(k-1\left|k-1\right.)A^{T}+Q$$
where $P(k-1\left|k-1\right.)$ indicates the covariance matrix forecasted by the state vector at k−1 time step, $P(k\left|k-1\right.)$ is the covariance matrix of the state vector at the k time step predicted by the state vector at the k−1 time step, $Q=q\left[\begin{array}{cc} \Delta t^{3}/3 & \Delta t^{2}/2\\ \Delta t^{2}/2 & \Delta t \end{array}\right]$ is the random noise covariance matrix in the discrete-time domain:

Once the displacement response at *t* time step is obtained, which can be expressed as
(19)$$z(i)=HX(i\left|i\right.-1)+d(i)$$
where z(i) means the observed displacement at time *t*, the observation matrix $H=\left[\begin{array}{cc} 1 & 0 \end{array}\right]$, d(i) represents measurement noise, assuming that the noise is a stationary zero mean Gaussian process, and the covariance matrix is $R=\frac{r}{\Delta t}$. In this paper, the maximum likelihood estimation method [22] is applied to calculate the noise parameters *q* and *r*. The state estimation vector $X(i\left|i\right.)$ and state estimation covariance matrix $P(i\left|i\right.)$ are
(20)$$X(i\left|i\right.)=X(i\left|i-1\right.)+K(i)(z(i)-HX(i\left|i-1\right.))$$
(21)$$P(i\left|i\right.)=(I-K(i)H)P(i\left|i-1\right.)$$
where $X(i\left|i-1\right.)$ means the state vector at the *i* time step forecasted by the state vector at the i−1 time step, $P(i\left|i-1\right.)$ is the covariance matrix of the state vector at the *i* time step predicted by the state vector at the i−1 time step, K(i) denotes the Kalman gain matrix at the *i* time step, which is calculated by the following formula
(22)$$K(i)=P(i\left|i-1\right.)H^{T}{\left(HP(i\left|i-1\right.)H^{T}+R\right)}^{-1}$$

### 2.2. Proposed Data Fusion Approach

The sampling rate of dynamic displacement derived from the above-mentioned mode superposition method is the same as that of strain, which is usually low in actual engineering. To make full use of the advantage of a high sampling rate of acceleration, an improved data fusion algorithm is developed to realize the information fusion between acceleration data and strain data to obtain high sampling rate displacement.

Firstly, the improved mode superposition method is used to derive displacement. The specific process is the same as above. In particular, unlike the existing mode superposition method that uses structural theory mode shape function, the proposed algorithm takes the SSI method to identify the strain mode, which has stronger applicability [23,24]. Therefore, the response generated when the bridge is subjected to steady state excitation is applied to identify the modal parameters [25].

After the strain mode shape of the structure is obtained, the displacement mode shape and modal coordinates can be calculated. Furthermore, the strain-derived displacement can also be acquired, which is abbreviated as $x_{u}$. Notably, the mode superposition method can be used to calculate the displacement response at any position after obtaining the strain response of the measurement point of the structure.

Then the strain-derived displacement $x_{u}$ is introduced in the above-mentioned Kalman filtering algorithm. Equations (17) and (18) are called the time update phase of Kalman filtering, and Equations (19)–(21) are the measurement update phase, which are the two main steps. In this proposed method, the time update phase is the same as before, while the change occurs in the measurement update phase. The observed displacement z(i) can be replaced by the strain-derived displacement $x_{u}$. Specifically, the observation equation during the measurement update process can be expressed as:(23)$$z(i)=x_{u}=HX(i\left|i\right.-1)+d(i)$$
where z(i) indicates the observed displacement, which is equal to the strain-derived displacement. Thus, the state estimation vector $X(i\left|i\right.)$ and state estimation covariance matrix in the measurement update process also has corresponding change:(24)$$X(i\left|i\right.)=X(i\left|i-1\right.)+K(i)(x_{u}-HX(i\left|i-1\right.))$$
where $X(i\left|i\right.)$ means the state vector at the *i* time step. $x_{u}$ is the strain-derived displacement. Finally, the reconstructed displacement in $x(i\left|i\right.)$ can be obtained. The framework of the proposed multi-rate data fusion algorithm is shown in Figure 2. It is worth noting that since the displacement and strain are in a one-to-one correspondence, the proposed method can reconstruct the dynamic displacement in real time.

## 3. Verification Using Simply Supported Beam

### 3.1. Finite Element Model

To validate the correctness and effectiveness of the proposed data fusion method, ANSYS software was applied to simulate the vibration of a simply supported beam under complex loads, which is composed of three different patterns, as shown in Figure 3; namely, a half-sine excitation with a pseudo-static component, a random excitation with a non-zero mean dynamic component, and an impulse excitation that can produce a free decay response. The beam188 element was utilized to simulate a simply supported beam; the length of the beam is 1.6 m, the cross-section is rectangular with 50×10 mm^2^, the elastic modulus is 206 GPa, the Poisson’s ratio is 0.3, and the density is set to 7800 kg/m^3^.

In the mode superposition method, the more mode orders are selected, the higher the accuracy of the reconstructed displacement. The first three-order modes have already accounted for most of the vibration energy. Considering that the first three-order modes of the structure can meet the requirements for a simply supported beam [26,27], four strain response extraction points were uniformly arranged in the length direction of the structure, and the middle span was set as the displacement and acceleration response extraction point. Excitation direction was Y and excitation duration was 20 s, the acceleration and displacement sampling rate were 1200 Hz, the strain sampling rate was 400 Hz, and the finite element model and measurement points layout are shown in Figure 4.

### 3.2. Displacement Reconstruction

Substituting the strain responses of the four measurement points into the state equation of the SSI method as state variables, a stability diagram can be obtained as shown in Figure 5. The orange curve in the figure represents the frequency spectrum, and the circle represents the point where the mode shape and frequency have stable solutions. The place where the peak of the frequency spectrum coincides with a row of circles can be considered as the excited strain mode, which can be obtained by reading the data corresponding to the circle, and the first three-order strain modes were successfully identified.

The identified first three-order strain modes are shown in Figure 6. To indicate the effectiveness of the SSI algorithm in identifying the strain mode shape, the theoretical displacement mode shape function of the simply supported beam was applied to obtain the theoretical strain mode shape function [28]; then, the mode identification method proposed by Wang et al. was used for comparison. Figure 7 compares the first-order strain modes identified by the different methods, which shows that the difference between the results identified by the SSI method and the theoretical solution is very small, while the method proposed by Wang et al. has a large identification error near the boundary. This is because the simulation in this paper only applies the load at the middle of the span. Therefore, the SSI method has great advantages in both the recognition accuracy and the operability.

In order to calculate the strain-derived displacement, the least square method is used to fit the mode shape curve to obtain the displacement mode function and modal coordinates separately.

Then, the identified third-order strain mode shape is fitted with a third-order polynomial function as the strain mode shape function of the structure, and then they are, respectively, integrated twice and substituted into the boundary conditions of the simply supported beam to obtain the corresponding displacement mode shape function. Moreover, the modal coordinates are acquired by substituting the strain mode function and strain response into Equation (2); then multiplying the displacement mode shape and modal coordinates together is the strain derived displacement. According to the proposed method, strain-derived displacement was used to correct the displacement obtained by acceleration integration can accurately reconstruct the dynamic displacement.

Figure 8 is the comparison diagram between the theoretical displacement extracted by ANSYS, the strain-derived displacement and reconstructed displacement by the data fusion method. Figure 8 shows that the strain-derived displacement curve, the displacement curve reconstructed by the data fusion algorithm, and the displacement curve extracted by ANSYS are in good agreement, which validates the correctness. The partially enlarged view suggests that the displacement reconstructed by the data fusion approach is more accurate than the strain-derived displacement, indicating that the proposed data fusion algorithm can not only increase the sampling rate of the strain-derived displacement but also make full use of the information contained in the acceleration to improve the accuracy of reconstructed displacement.

For comparing the performance between the proposed method and the similar existing approach, the method developed by Zhu et al. [29] was used to reconstruct the dynamic displacement at the same position. Due to the limitation of the strain-derived displacement calculated by the method of Zhu et al., the number of strain measurement points needs to be increased to 10, which are 0.01, 0.64, 0.8, 0.96, and 1.6 m away from the left support, respectively, and it is required to extract strain on the upper and lower surfaces of each measurement point simultaneously. The displacement calculated by this method is compared with that reconstructed by the proposed approach, as shown in Figure 9. It can be seen that the displacements calculated by the two methods are in good agreement with the theoretical displacement, and the partial enlarged view shows that the accuracy of the proposed method is higher even with fewer measurement points. In addition, the method of Zhu et al. requires that strain measurement points should be arranged at specific locations such as the structural boundary and the reconstruction target points, and each measurement point needs to be equipped with sensors on both sides of the structure, which undoubtedly limits the popularization and application of this method in actual engineering.

### 3.3. Parametric Analysis

With the purpose of applying the proposed algorithm to actual engineering, further analysis of its robustness is needed. Moreover, the normalized root mean square error (NRMS) is defined to measure the accuracy of reconstructed displacement:(25)$$\mathrm{NRMS}=\frac{\sqrt{\frac{1}{N}\sum_{n=1}^{N}{\left(d_{r}-d_{t}\right)}^{2}}}{\sqrt{\sum_{n=1}^{N}{\left(d_{r}\right)}^{2}}}\times 100\%$$
where *N* is the total number of data points, $d_{r}$ is the reconstructed displacement value, and $d_{t}$ represents the theoretical displacement value.

The anti-noise performance of the method is an important indicator of its applicability. Signal noise level is usually measured by signal-to-noise ratio (*SNR*), defined as
(26)SNR=10log(S/N)
where *S* is the average power of the signal, and *N* is the average power of the noise. It can be seen that higher *SNR* indicates less noise contained in the signal.

The acceleration and strain responses extracted by ANSYS were added with Gaussian white noise with a signal to noise ratio of 5–100 dB, and then the data fusion algorithm was used to reconstruct the displacement. Figure 10 shows displacement time histories with a signal to noise ratio of 5 dB. The reconstruction accuracy is reduced compared with Figure 8, and the error of pseudo-static displacement increases most obviously, while the reconstruction of pseudo-static displacement mainly depends on strain-derived displacement, which indicates that the mode superposition method is easily affected by noise. The reconstruction effect of high-frequency components of the displacement is better, showing that the reconstruction of high-frequency displacement is more dependent on acceleration, and the data fusion algorithm can sufficiently reduce the influence of noise. Overall, the NRMS is still only 5.43%, illustrating that the mode superposition method is not as good as the data fusion method in resisting noise.

Table 1 shows the NRMSs of the reconstructed displacement after adding different signal to noise ratio noise to the dynamic response, which shows that NRMS increases as the *SNR* decreases, but even if the *SNR* is only 5 dB, the NRMS is only 5.43%, indicating that the proposed data fusion algorithm has good anti-noise performance.

In actual engineering, the acceleration sampling rate is usually higher than that of the strain. Consequently, it is necessary to analyze the accuracy of the reconstructed displacement of the data fusion method in the case of different sampling rate ratios. Define the sampling rate ratio as $Scale=f_{a}/f_{s}$, where $f_{a}$ is the acceleration sampling rate and $f_{s}$ is the strain sampling rate. $f_{a}$ was set to 1200 Hz, and $f_{s}$ was set to 400, 60, and 24 Hz; that is, the sampling rate ratio *Scale* was 3, 20, and 50, respectively. The *SNR* of Gaussian white noise was still 100 dB. Figure 11 shows the displacement time histories when *Scale* = 50. Compared with Figure 8, the degree of agreement between the two curves is decreased. This is because the increase in the ratio of sampling rates results in a longer measurement update interval, which in turn increases the error. Furthermore, the pseudo-static displacement error is larger than that of the high-frequency displacement; but in fact, the reduction in the strain sampling rate will not affect the reconstruction of the pseudo-static displacement since the strain corresponding to the pseudo-static displacement changes slowly. On the contrary, the reconstructed displacement accuracy of the high-frequency displacement is more limited by the strain sampling rate due to the rapid change in the corresponding strain. This situation occurs owing to the noise added in the acceleration and strain response is unreasonable.

Figure 12 is a comparison diagram of displacement time histories when the signal to noise ratio of acceleration and strain noise is different. This figure shows that the pseudo-static displacement error is small, and the high-frequency displacement error is large, which conforms to the previous judgment. Nevertheless, the parameter analysis is mainly aimed at the robustness of the algorithm, so the acceleration and strain noise levels remain consistent in the subsequent discussion.

Table 2 presents the NRMSs corresponding to the different sampling rate ratios. It can be seen that as the *Scale* increases, NRMS also rises, which is because the displacement derived from the strain is needed to guide the correction process of the displacement in the correction phase of the Kalman filtering, the time interval of the correction phase will also become longer when the strain sampling rate is reduced, resulting in larger reconstruction errors. Even when the *Scale* reaches 50, the NRMS is still only 2.7%, which shows that the proposed data fusion algorithm still has good robustness when the *Scale* is large.

Different displacement reconstruction points may also be encountered in actual projects, and the signal to noise ratio of different measurement point positions when collecting signals is also different. Hence, ANSYS is used to extract acceleration and displacement responses at different positions for analysis and research. Figure 13 is a comparison diagram between reconstructed and theoretical displacements at a distance of 0.2 m from the left support. The reconstruction accuracy of pseudo-static displacement and high-frequency displacement is slightly reduced as the measurement point is relatively close to the mode zero point of the first-order displacement mode of the structure, and the error of obtaining the displacement mode shape from the strain mode shape increases, which leads to a larger error of the strain-derived displacement.

Table 3 shows the NRMSs at different locations, which shows that the errors at 0.4, 0.6, and 0.8 m are very close except for the slight increase in NRMS at 0.2 m. Still, when the distance to the left support is only 0.2 m, the error is only 1.61%, indicating that the proposed data fusion algorithm can effectively reconstruct the displacement at a position closer to the left support.

In addition, we also compared the time required for the proposed method to process data of different durations (the sampling rate is 1200 Hz), and the results are shown in Table 4. Obviously, the data processing time is far less than the data duration; therefore, the proposed method will not cause data stack and can achieve online reconstruction.

## 4. Validation of Complex Beam Structures

### 4.1. Model Description

In order to verify the reliability of the proposed method for complex beam structures, finite element models of a two-span continuous beam and a beam with a non-uniform cross-section were established and analyzed. The model size and sensor layout are shown in Figure 14 It is worth noting that since the reconstructed displacement target point of the continuous beam is located on the right span, the sensors are only placed on the right span.

The form of excitation is the same as that of the excitation in Section 3, but the amplitude is different. The ANSYS software was applied to calculate the dynamic response of the two structures. The strain sampling rate is 400 Hz, and the acceleration and displacement sampling rate are 1200 Hz. No noise was added to the response to clearly verify the correctness of the proposed method.

### 4.2. Displacement Reconstruction Results

The SSI method was used to process strain data and obtain strain mode shape. The right span of the continuous beam was taken as a single simply supported beam to calculate the displacement mode and strain-derived displacement by the mode shape superposition method. Simultaneously, the segment mode shape superposition method was applied to the non-uniform beam. The calculated displacement mode shape was compared with the ANSYS modal analysis result, as shown in Figure 15. It can be seen that the displacement mode shape of the continuous beam target span and the non-uniform beam is quite different from the theoretical solution of the mode shape of the uniform simply supported beam, but the proposed method can still accurately calculate the two displacement mode shapes.

The proposed data fusion method was utilized to reconstruct the dynamic displacements of the above two structures, and the results are shown in Figure 16. It can be seen from the figure that the proposed method can accurately reconstruct the dynamic displacement of the complex beam structure and has good practical value.

## 5. Experiment on a Simply Supported Beam

### 5.1. Experimental Setup

For further study of the application of the method in the actual structure, the steel material was processed into a simply supported beam model for model experiments. The simply supported beam model is 1.6 m long, has a section size of 49.5×10 mm^2^, an elastic modulus of 210 GPa, and a Poisson’s ratio of 0.3. Five resistance strain gauges as well as one piezoelectric acceleration sensor were uniformly arranged along the length of the beam to measure the strain and acceleration response, respectively. A laser displacement sensor was applied to measure the displacement response. The sensor layout is shown in Figure 17.

Excitation was applied on the simply supported beam model in the vertical direction at the mid-span position, and the loading method adopted the manual push method to produce pseudo-static and high-frequency displacement components. The acceleration and strain sensors were demodulated by the photoelectric synchronous demodulator independently developed by the research group, and the signal of the laser displacement meter was collected by the Donghua dynamic signal acquisition and analysis system. The acceleration sampling rate was set to 1200 Hz, the strain sampling rate was set to 400 Hz, the displacement sampling rate was set to 10 kHz, and the acquisition time was 20 s. In the later data processing, the method of resampling is used to reduce the displacement sampling rate to 1200 Hz. The built experimental platform is shown in Figure 18.

### 5.2. Displacement Reconstruction

The mode superposition method and the multi-rate Kalman filtering algorithm were used to process the strain response and acceleration response, and the reconstructed displacement was obtained and compared with the data measured by the laser displacement sensor. The result is shown in Figure 19, which demonstrates that the displacement curve reconstructed by the data fusion method and the strain-derived displacement curve have a high degree of overlap with the measured displacement curve. The pseudo-static displacement part is almost completely coincident, which shows that the reconstruction of the pseudo-static displacement mainly depends on the accuracy of the strain-derived displacement. The partial enlargement illustrates that the accuracy of the high-frequency displacement part derived from the strain is slightly reduced, but the data fusion algorithm can effectively improve the precision of this part, indicating that the high-frequency displacement information contained in the acceleration is more than that of the strain.

### 5.3. Results Analysis

For the purpose of further verifying the performance of the algorithm for actual structures, it is necessary to analyze its robustness. The anti-noise performance of the method is no longer analyzed here since the data collected in the model experiment already contains a certain level of noise, and the case of different sampling rate ratios is studied first. The acceleration sampling rate was set to 1200 Hz, and the strain sampling rates were 400, 120, 60, and 24 Hz; that is, the *Scale* was 3, 10, 20, and 50, respectively.

Figure 20 displays the displacement reconstructed by different Kalman filtering algorithms when *Scale* = 50. Compared with Figure 19, the accuracy of the reconstructed displacement is decreased; this is since the increase in the sampling rate ratio leads to a longer measurement update interval, which in turn increases the error. The degree of overlap between the displacement curve and the measured displacement curve is still high. The NRMS is only 7.29%, which shows that the proposed data fusion method is suitable for working conditions with a low strain sampling rate.

Table 5 shows the NRMSs corresponding to different *Scales*. The displacement reconstruction error rises with the increase in the sampling rate ratio, which is the same as the previous numerical simulation section. The reason is also that the correction process of Kalman filtering requires the guidance of strain-derived displacement, and the decrease in the strain sampling rate leads to less guidance process, which leads to increased error. However, when the sampling rate ratio reaches 50, the NRMS is only 7.29%, indicating that the algorithm is suitable for low sampling rate strain.

To analyze the performance of the proposed algorithm when applied to different measurement points, vibration experiments were carried out at 0.2, 0.4, and 0.6 m from the left support, respectively. Figure 21 is the displacement history at 0.2 m. Compared with Figure 19, the reconstruction accuracy of high-frequency and pseudo-static displacement is all slightly reduced, which is similar to the simulation section and the reason is also that the measurement point is close to the zero point of the first-order mode. Still, the NRMS is only 6.11%, which proves that the fusion method is suitable for positions close to the support.

Table 6 demonstrates the NRMSs at different locations. It can be seen from the table that the errors at 0.4, 0.6, and 0.8 m are all around 5% except for the slight increase in NRMS at 0.2 m. The closeness of the error is not as good as the numerical simulation section, because the excitation between multiple experiments is different, and the displacement amplitude at the same measurement point is also different, while it does not affect the algorithm performance analysis. When the distance from the left support is only 0.2 m, the NRMS is only 6.11%, indicating that the proposed data fusion algorithm has strong applicability.

## 6. Summary and Conclusions

This paper proposed a smart multi-rate data fusion method using acceleration and strain data for dynamic displacement reconstruction, which can reconstruct structural dynamic displacement online. Different from the existing method of displacement derived by strain, the approach first used the SSI method to process the strain response of several measurement points to obtain the strain mode shape of the structure and then utilized the mode superposition method to acquire the strain-derived displacement. Finally, the acceleration and strain-derived displacement were combined as input values in the developed multi-rate Kalman filtering algorithm, which was applied to increase the sampling rate and accuracy of the strain-derived displacement, and it was verified by numerical simulation and model experiment of the simply supported beam. The conclusions are as follows:

(1) This method can still maintain good performance when the acceleration and strain sampling rates differ greatly. The NRMS is 2.7% when the sampling rate ratio is 50 in the numerical simulation section; this value is 7.29% in the model experiment.

(2) The proposed approach has good anti-noise performance. The NRMS is only 5.43% when noise with a signal to noise ratio of 5 dB is added to the acceleration and strain responses in the numerical simulation. The measurement data with noise in the experiment can also reconstruct the displacement well.

(3) The signal to noise ratio of the measurement is not high because it is close to the zero point of the first-order mode at a place close to the support. However, this algorithm still shows strong applicability. The NRMS corresponding to 0.2 m from the support in the numerical simulation is 1.61%, and the value in the experiment is 6.11%.

(4) The reconstructed displacement and the actual displacement are very close in the time domain. In addition, the error of reconstructing the displacement of continuous beam and non-uniform beam by the proposed method is only 1% and 2%, respectively.

The proposed data fusion method can comprehensively utilize the collected acceleration and strain responses to realize the complementary advantages of the two, and provide an effective method for indirect acquisition of structural dynamic displacement, which realize the fusion of the SSI method, mode shape superposition approach, and Kalman filtering algorithm. The comparison with existing methods also shows that the proposed method has stronger advantages. Numerical simulations of continuous beam and non-uniform beam prove that the proposed method is also suitable for complex beam structures. It is believed that this method has broad application prospects in the field of bridge structural health monitoring.

At present, the proposed method can only reconstruct the displacement at the measuring point where the acceleration sensor located and cannot obtain the full-field displacement of the structure. In addition, the simply supported beam model is used to verify the proposed theory of this paper, while the actual bridge structure is far more complicated than the used model. Thus, the next step should focus on the application in practical engineering, making it suitable for complex structures.

## Figures and Tables

**Figure 1 sensors-22-03167-f001:**
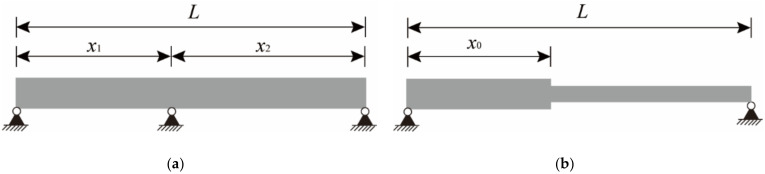
Complex beam structures. (**a**) Two span continuous beam. (**b**) Non-uniform simply supported beam.

**Figure 2 sensors-22-03167-f002:**
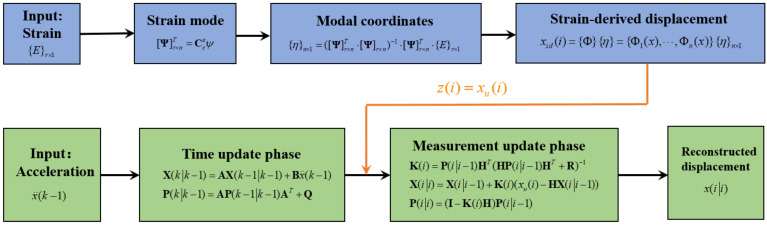
Multi-rate data fusion algorithm framework.

**Figure 3 sensors-22-03167-f003:**
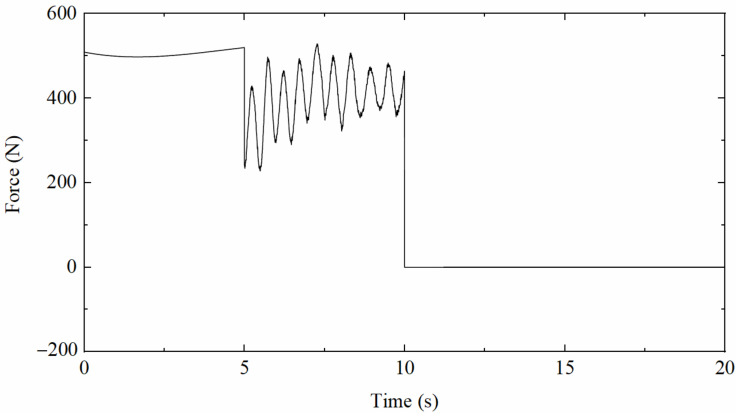
Time history of the applied excitation.

**Figure 4 sensors-22-03167-f004:**
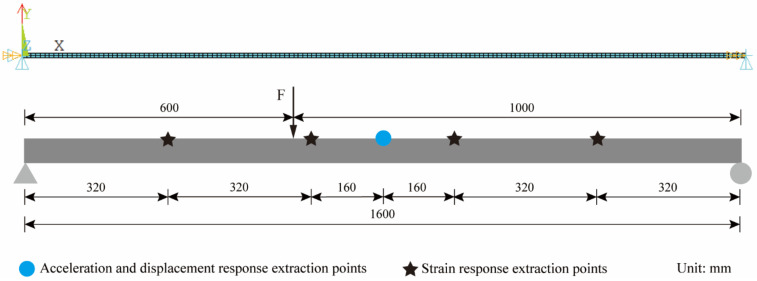
Finite element model and measurement points layout.

**Figure 5 sensors-22-03167-f005:**
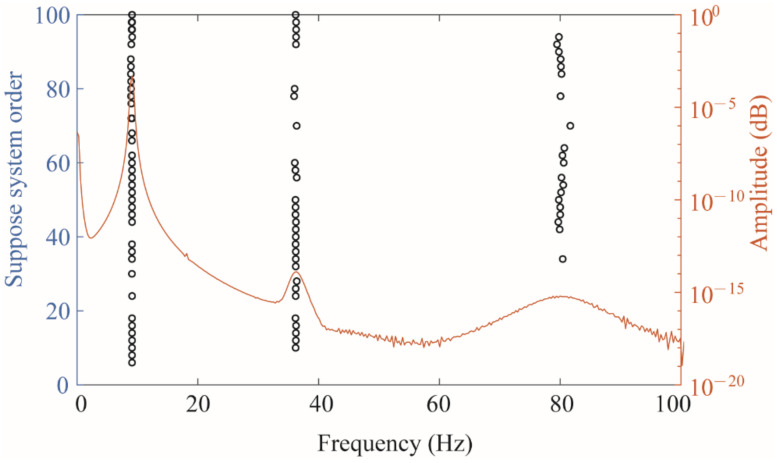
The stability diagram with a spectrum diagram.

**Figure 6 sensors-22-03167-f006:**
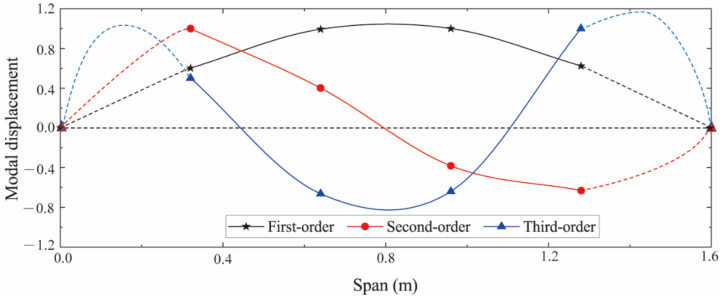
First three-order strain modes identified by SSI method.

**Figure 7 sensors-22-03167-f007:**
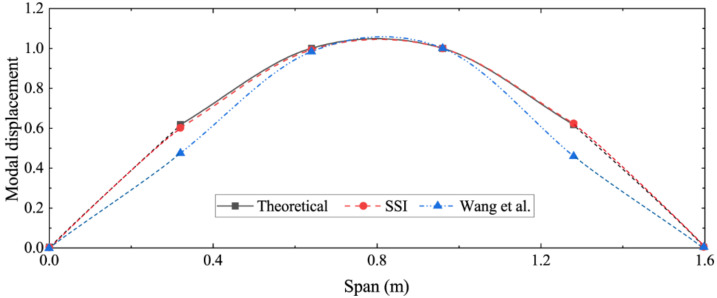
Comparison of first-order strain modes identified by different methods.

**Figure 8 sensors-22-03167-f008:**
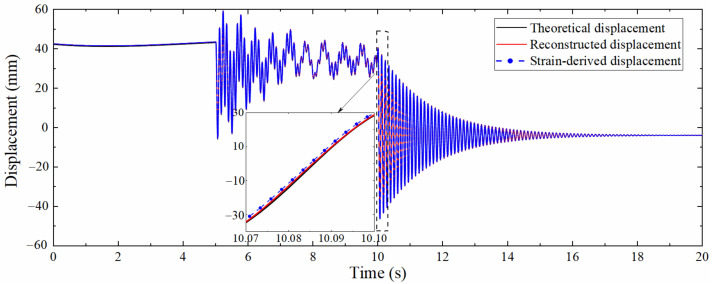
Comparison between theoretical displacement and reconstructed displacement.

**Figure 9 sensors-22-03167-f009:**
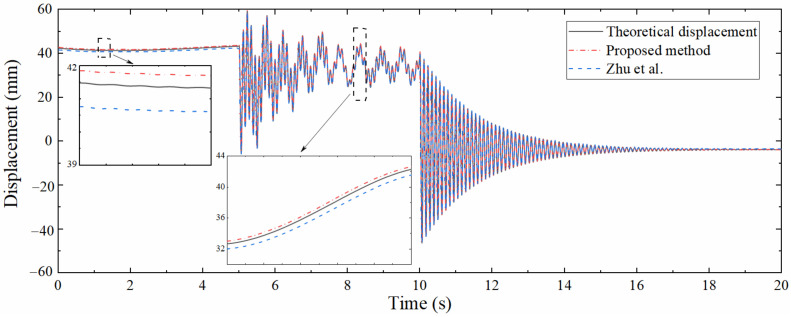
Performance comparison between proposed method and approach of Zhu et al.

**Figure 10 sensors-22-03167-f010:**
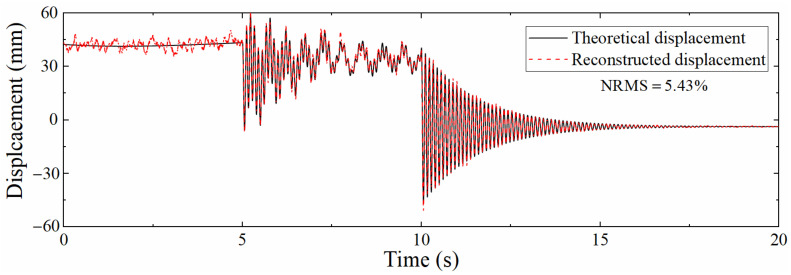
Comparison between theoretical displacement and reconstructed displacement (5 dB).

**Figure 11 sensors-22-03167-f011:**
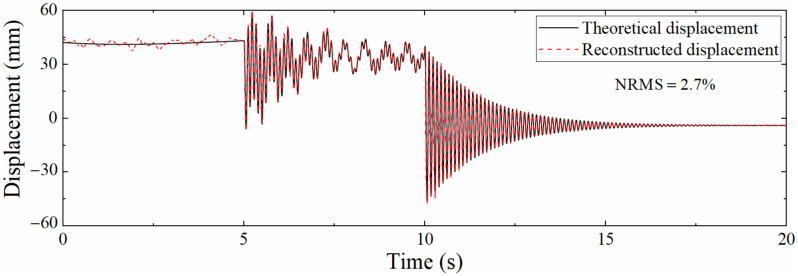
Comparison between theoretical displacement and reconstructed displacement (*Scale* = 50).

**Figure 12 sensors-22-03167-f012:**
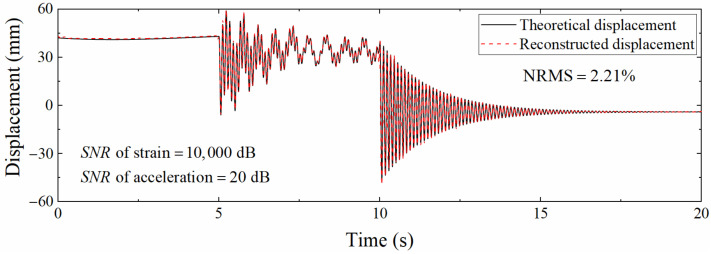
Comparison of different displacements at different signal to noise ratios (*Scale* = 50).

**Figure 13 sensors-22-03167-f013:**
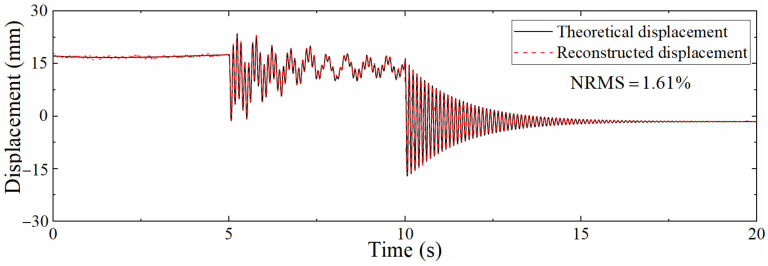
Comparison between theoretical displacement and reconstructed displacement time history (0.2 m from the left support).

**Figure 14 sensors-22-03167-f014:**
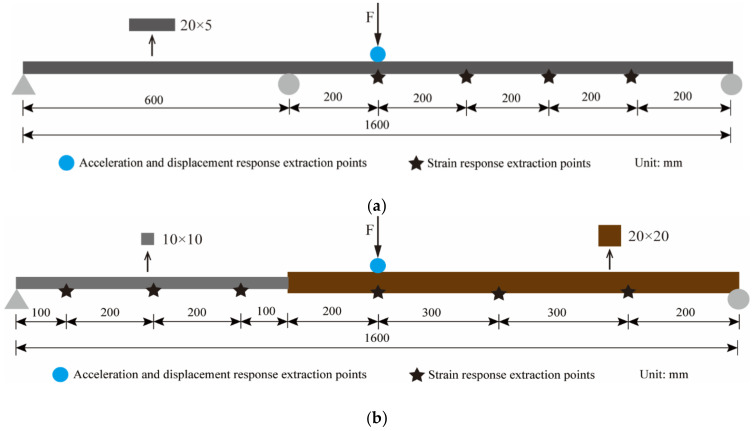
Sensor layout and model size. (**a**) Continuous beam. (**b**) Simply supported beam with non-uniform section.

**Figure 15 sensors-22-03167-f015:**
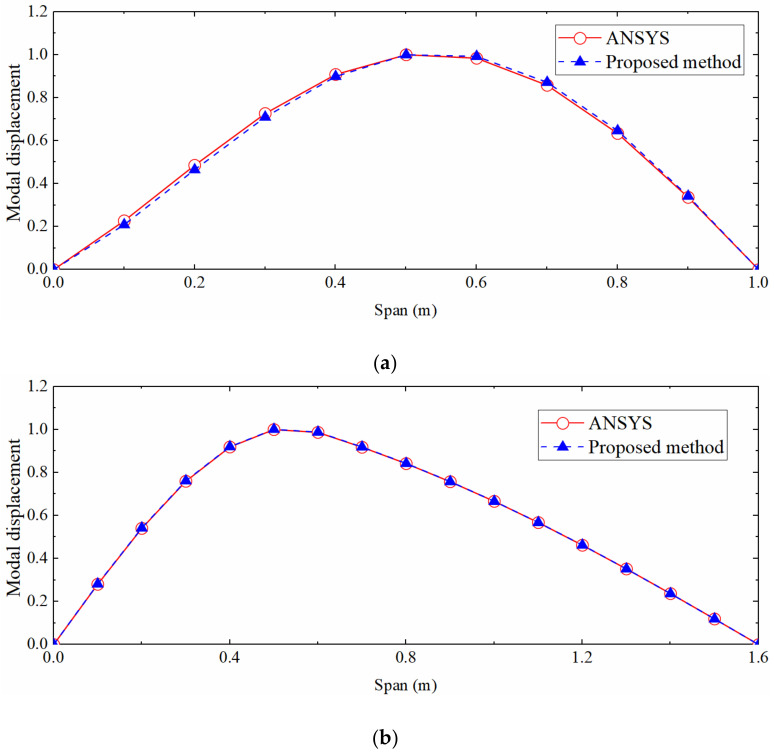
Comparison of first-order displacement mode shapes. (**a**) Continuous beam; (**b**) Simply supported beam with non-uniform section.

**Figure 16 sensors-22-03167-f016:**
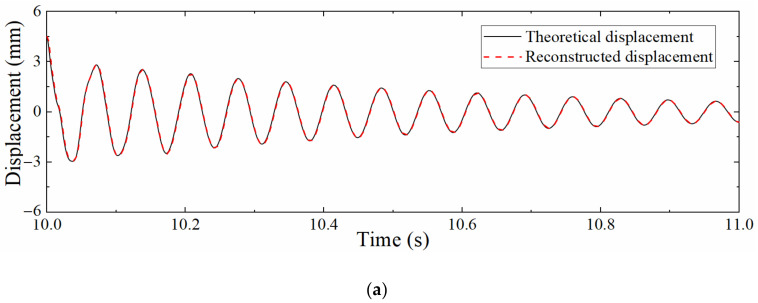
Displacement reconstruction result of complex beam structures. (**a**) Continuous beam; (**b**) Simply supported beam with non-uniform section.

**Figure 17 sensors-22-03167-f017:**
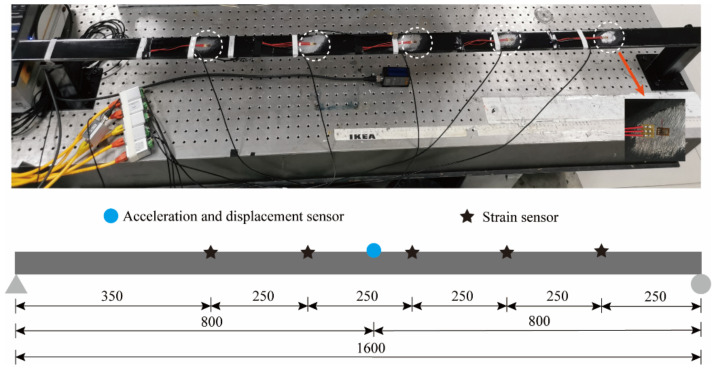
Sensor layout (the unit of length in this figure is mm).

**Figure 18 sensors-22-03167-f018:**
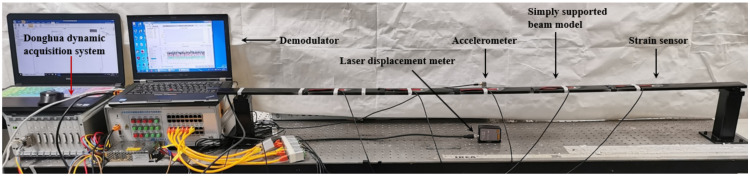
Simply supported beam vibration experiment system.

**Figure 19 sensors-22-03167-f019:**
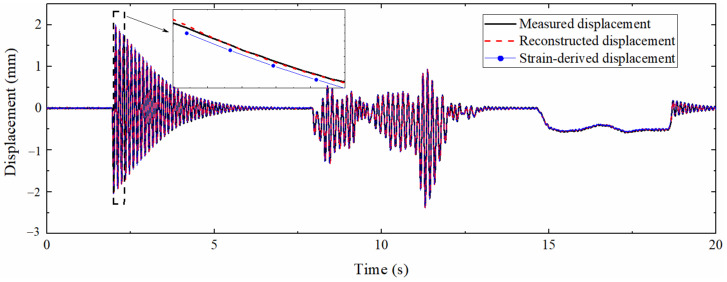
Comparison between measured displacement and reconstructed displacement.

**Figure 20 sensors-22-03167-f020:**
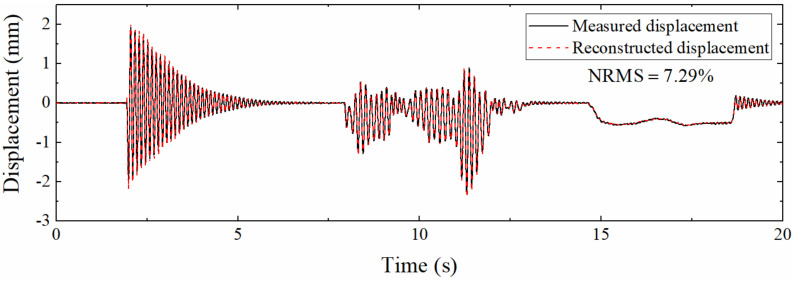
Displacement reconstructed by the proposed algorithm (*Scale* = 50).

**Figure 21 sensors-22-03167-f021:**
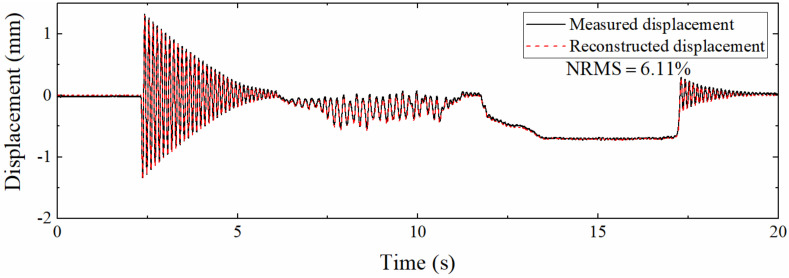
Comparison between different displacements (0.2 m from the left support).

**Table 1 sensors-22-03167-t001:** NRMSs corresponding to different signal to noise ratios.

Signal to Noise Ratio (dB)	NRMS (%)
5	5.43
20	3.04
50	2.09
100	1.49

**Table 2 sensors-22-03167-t002:** NRMSs corresponding to different *Scales*.

*Scale*	NRMS (%)
3	1.49
20	2.09
50	2.71

**Table 3 sensors-22-03167-t003:** NRMSs corresponding to different locations.

Distance from Left Support (m)	NRMS (%)
0.2	1.61
0.4	1.31
0.6	1.34
0.8	1.49

**Table 4 sensors-22-03167-t004:** Comparison between data duration and calculation duration.

Data duration (s)	20	40	60
Processing time (s)	3.2	4.1	5.4

**Table 5 sensors-22-03167-t005:** NRMSs corresponding to different *Scales*.

*Scale*	NRMS (%)
3	3.93
10	4.25
20	4.79
50	7.29

**Table 6 sensors-22-03167-t006:** NRMSs corresponding to different locations.

Distance from Left Support (m)	NRMS (%)
0.2	6.11
0.4	5.43
0.6	4.98
0.8	3.93

## Data Availability

All data, models, and code generated or used during the study appear in the submitted article.

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
