# Peer review of "A Smart Multi-Rate Data Fusion Method for Displacement Reconstruction of Beam Structures"

_sensors, 2022, doi:10.3390/s22093167_

Round 1

Reviewer 1 Report

The authors proposed a multi-rate data fusion technique for displacement reconstruction of beam structures. The manuscript is well-written and adequately presented. The reviewer has the following concerns:

  1. The authors mention the reconstruction error is 5% for the noise level of 5dB. Did the authors check the proposed method for noise levels less than 5dB? The trend shows the higher the signal-to-noise ratio, the less error. Clarify?
  2. Why did the authors choose specifically SSI for mode shapes?
  3. The authors should add disadvantages of the proposed method too to improve the readability of the article. 
  4. Did the authors validate the proposed method on any real-life structure? Why or Why not?

Reviewer 2 Report

This manuscript presents a novel method for dynamic displacement reconstruction of beam structures in a real-time manner. The effectiveness of the method was demonstrated with both numerical and experimental tests with less than 5% error. 

The manuscript is very well-written and presented. It can be accepted for publication by Sensors after considering the following minor comments: 

  1. As this study aimed to achieve online reconstruction, can you elaborate on the online real-time performance for both numerical and experimental validations?
  2. Line 138, full term of abbreviations such as SSI here, should be given when first appear.
  3. Line 247, can you be more specific about what "complex" excitation pattern was applied?

Round 2

Reviewer 1 Report

The authors have clarified and improved the manuscript.